# Mitogenome of the leaf-footed bug *Notobitus montanus* (Hemiptera: Coreidae) and a phylogenetic analysis of Coreoidea

**Xiaoke Tian, Yongqin Li, Qin Chen, Qianquan Chen** * 

School of Life Sciences, Guizhou Normal University, Gui'an, Guizhou, China

* qqchen@gznu.edu.cn

## Abstract

*Notobitus montanus* Hsiao, 1963 is a major pest of bamboos. The mitogenome of *N. montanus* (ON052831) was decoded using next-generation sequencing. The mitogenome, with 42.26% A, 30.54% T, 16.54% C, and 10.65% G, is 16,209 bp in size. Codon usage analysis indicated that high frequently used codons used either A or T at the third position of the codon. Amino acid usage analysis showed that leucine 2, phenylalanine, isoleucine and tyrosine were the most abundant in 31 Coreoidea species. Thirteen protein-coding genes (PCGs) were evolving under purifying selection, *nad5* and *cox1* had the lowest and strongest purifying selection stress, respectively. Correlation analysis showed that evolutionary rate had positive correlation with A+T content. No tandem repeat was detected in the non-coding region of *N. montanus*. The phylogenetic tree showed that Alydidae and Coreidae were not monophyletic. However, the topology of phylogenetic trees, based on 13 PCGs, was in accordance with that of tree based on both mitochondrial and nuclear genes but not ultraconserved element loci or combination of 13 PCGs and two rRNAs. It seems that their relationships are complex, which need revaluation and revision. The mitogenomic information of *N. montanus* could shed light on the evolution of Coreoidea.

## Introduction

Leaf-footed bugs, hemipteran superfamily Coreoidea, are phytophagous insects which consist of several forest and agricultural pests [1, 2]. For example, *Notobitus montanus* Hsiao, 1963 is a major pest of bamboos, which not only supply human with foods, building materials, crafts, and high-quality paper, but also are involved in landscaping and soil conservation [3]. It distributes in many southern provinces of China, such as Zhejiang, Sichuan, Yunnan, Guizhou, which serves as the pest of serval species of bamboo, including, *Phyllostachys sulphurea* var. Viridis, *Phyllostachys glauca*, *Phyllostachys heteroclada*, *Phyllostachys bambusoides* f. shouzhu, *Phyllostachys heteroclada* var. pubescens, *Pleioblastus amzrus*, *Phyllostachys praecox* f. provernalis [4]. The adults and nymphs of *N. montanus* feed on the sap of bamboo shoot and young bamboo, making bamboo falls into a decline and even dead. A female can lay 20–70 eggs, and the egg hatchability is high. Eggs take up 15–20 days to develop as nymphs, and then nymphs take up 30–50 days to develop as adult, which can live for 330–350 days [4]. Due to its high

**Data Availability Statement:** The genome sequence data that support the findings of this study are openly available in GenBank of NCBI at

(https://www.ncbi.nlm.nih.gov/) under the accession no. ON052831.

**Funding:** This research was funded by National Natural Science Foundation of China, grant number 32060124; Guizhou Normal University, grant number Qianshixinmiao[2021]A11 The funders had no role in study design, data collection and analysis, decision to publish, or preparation of the manuscript.

**Competing interests:** The authors have declared that no competing interests exist.

reproduction, strong ability of flying and gregariousness, *N. montanus* can cause serious economic loss of bamboo [5]. For example, in 1973, 90% of bamboo shoots were attacked by it in Liangping count, Sichuan province, leading 16.4% bamboo shoots dead [4]. Currently, approximately 30 Coreoidea species are recorded as the pest of bamboo.

In addition to two extinct families (Trisegmentatidae and Yruipopovinidae), Coreoidea consists of five families: Coreidae (2571 species), Alydidae (282 species), Rhopalidae (224 species), Stenocephalidae (30 species) and Hyocephalidae (three species) [1, 6, 7]. The Coreidae is divided into four subfamilies: Coreinae, Hydarinae, Meropachyinae, and Pseudophloeinae [6, 7]. Coreinae, with 2,320 (90%) species (372 genera), is the largest subfamily of Coreidae. In the past, the taxonomic classification of Coreidae was mainly based on morphological traits such as apical spine or tooth on the hind tibiae, hind femora, and metathoracic scent gland orifices [8]. The phylogenetic relationships among the tribal rank taxa of Coreidae have not been investigated comprehensively since 1997 [8, 9]. However, some studies indicated that these morphological traits exhibited homoplasy [1, 9], which might explain contradictory results from different studies [1]. Phylogenomic trees constructed with ultraconserved element loci and mitogenome showed that both Coreidae and Alydidae were not monophyly [1, 6]. Furthermore, the phylogenomic analysis showed that several genera, and subfamilies of Coreidae were para- and polyphyly, which suggested that the taxonomic classification of Coreidae need revaluation and revision [9]. In short, leaf-footed bugs' phylogenetic relationships have remained far from solved [1].

Animal mitochondrial genome (mitogenome) is a circular double-stranded DNA molecule. Generally, it encodes 13 protein-coding genes (PCGs), two ribosomal RNA (rRNAs), and 22 transfer RNA (tRNAs) [10]. In addition, it contains a non-coding control region (A+T-rich region, D-loop), which contains essential regulatory elements for replication and transcription [11]. Owing to the lack of protection of histones, mitogenomic DNA has a much higher mutation frequency than nuclear DNA. As a result, mitogenomes are widely used in phylogenetics, phylogeography, population genetics, and evolutionary biology [12, 13]. With the development of next-generation sequencing (high-throughput sequencing), many mitogenomes have been decoded using this technique.

In this work, the mitogenome of *N. montanus* has been decoded. The characteristics of the mitogenome, including nucleotide composition, codon usage, tRNA secondary structure, and the evolutionary pattern of 13 PCGs, were systematically analyzed. The phylogenetic tree of Coreoidea was constructed with the new mitogenome and 30 mitogenomes extracted from NCBI (S1 Table). The mitogenome could shed light on the evolution of Coreoidea.

## Materials and methods

### Samples and identification

*Notobitus montanus* is a common pest of bamboos in China as well as not recorded in the species list-of-ethics committees for research involving animals of the Guizhou Normal University. Therefore, no ethical approval or other relevant permission can be provided for the study. Specimens of *N. montanus* were collected from the campus of Guizhou Normal University (26°22′50.30″N, 106°38′11.72″E) in July 2021. All specimens were identified from their morphological characteristics. Total DNA was then extracted from the muscle tissue of an adult specimen with the phenol-chloroform extracting method [14, 15]. The fragment of mitochondrial cytochrome c oxidase subunit I gene (*cox1*) was amplified with primers (LCO1490 and HCO2198) [16]. The PCR productions were checked by agarose gel electrophoresis and sequenced by sanger sequencing. The sequence was submitted to BOLD systems v4 (http://www.boldsystems.org/) as a query for species identification. The similarity between the query

sequence and reference sequence of *N. montanus* in the BOLD was 99.77%. The specimens (specimen ID: GZNU-cqq-136) were soaked in absolute alcohol and then stored at 4˚C in the Museum of Guizhou Normal University.

### Next-generation sequencing, annotation, and bioinformatics analysis

Total DNA was isolated from the muscle tissue of an adult specimen with ONE-4-ALL Genomic DNA Mini-Prep Kit (BS88504, Sangon, Shanghai, China). DNA was fragmented, then ~500 bp DNA was recycled. Paired-end libraries were constructed with the Illumina platform. The DNA was sequenced using the Illumina Hiseq X Ten at the Sangon Biotechnology Company (Shanghai, China). The adapter sequences were removed, and low-quality reads were trimmed with Trimmomatic version 0.36 [17]. The mitogenome of *Cloresmus pulchellus* (NC_042806) was used as reference, and clean reads were assembled with SOAPdenovo2 (version 2.04) [18]. PCGs were identified by BLAST comparison with *C. pulchellus* mitogenome [19]. The secondary structures of tRNAs were predicted with MITOS2 and tRNAscan-SE 2.0 [20, 21], rRNAs and non-coding control region were determined by the boundary of tRNAs. Nucleotide composition was calculated with MEGA X [22]. The AT-skew values were calculated by the following formula: AT-skew = (A—T)/(A + T). Similarly, GC-skew values were calculated by GC-skew = (G—C)/(G + C). Codon usage indexes, including codons counts, and relative synonymous codon usage (RSCU), were calculated with MEGA X [22]. The number of nonsynonymous substitutions per nonsynonymous site (Ka), and the number of synonymous substitutions per synonymous site (Ks) were calculated with DnaSP 6 [23]. Tandem repeats in the non-coding control region were identified using tandem repeats online finder server [24]. Circular map of the *N. montanus* mitogenome was generated with CGView server [25]. Heatmap of codon and amino acid usage was generate with ggplot2 as implemented in R v4.1.2. Figures were edited with Adobe Illustrator CS5.

### Phylogenetic analysis

The mitogenomes of *N. montanus* and 30 available complete mitogenomes of Coreoidea in NCBI were used to construct the phylogenetic tree of Coreoidea (S1 Table). *Malcus inconspicuus* (Hemiptera: Malcidae), *Physopelta gutta* (Hemiptera: Largidae), and *Nezara viridula* (Hemiptera: Pentatomidae) were selected as representative of the outgroups. These mitogenomes were imported into PhyloSuite V1.2.2 [26]. Then the nucleic acid sequences of 13 PCGs were extracted from these mitogenomes. Codon-based multiple alignments were carried out with MAFFT as implemented in PhyloSuite [26]. Then alignments of 13 PCGs were concatenated. PartitionFinder2 was used to select the best-fit partitioning strategy and models for the concatenated sequences. Phylogenetic trees were reconstructed by the Bayesian (Mrbayes v 3.2.6) and maximum likelihood (IQ-TREE v1.6.8) method as implemented in PhyloSuite [26]. The number of generations was 10, 000, 000 and Partition Models were selected as models. The phylogenetic tree was visualized with Figtree v1.4.4 and Adobe Illustrator CS5.

## Results

### Genomic structure and nucleotide composition

The complete mitogenome of *N. montanus* (ON052831) was 16209 bp, which encoded 13 PCGs, 22 tRNAs, two rRNAs, and a non-coding control region (D-loop) (Fig 1 and S2 Table). Fourteen genes, including four PCGs, eight tRNAs, and two rRNAs, were encoded by the minority strand (N strand), and the remaining genes were encoded by the majority strand (J strand). The gene arrangement of *N. montanus* was in accordance with that of other Coreoidea

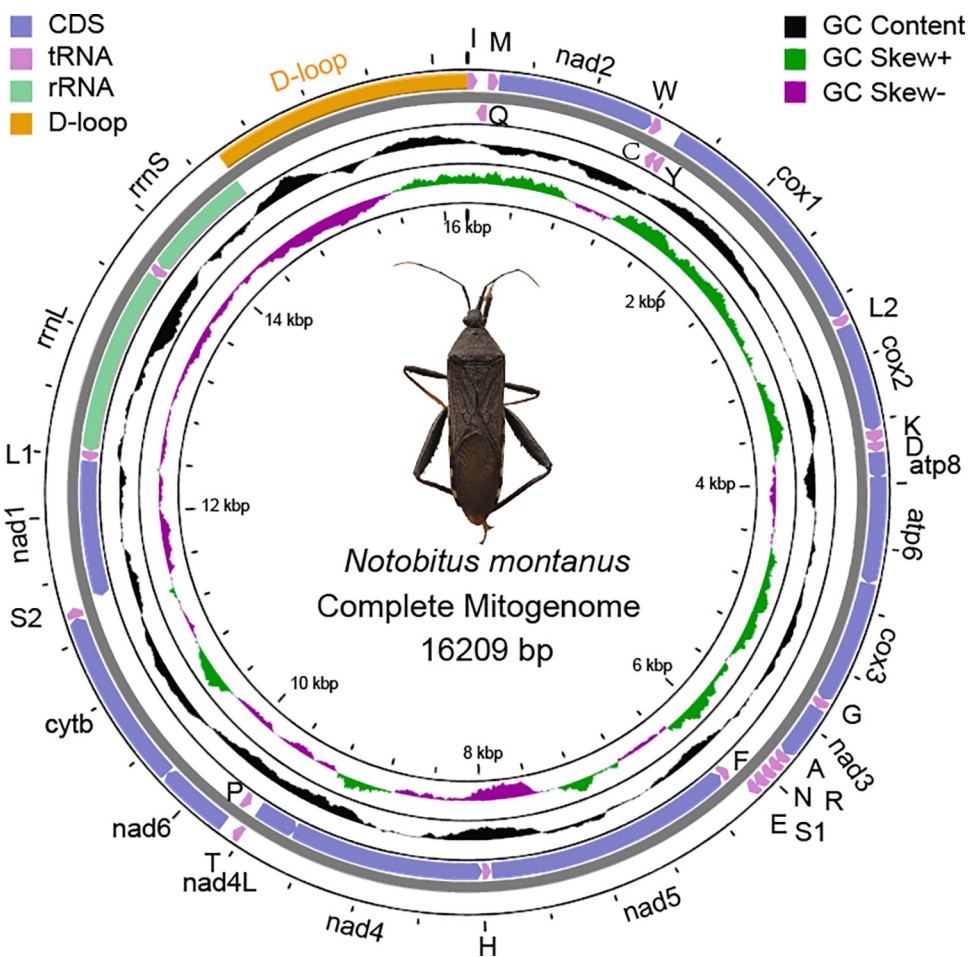

**Fig 1. Mitogenome map of *Notobitus montanus*.** Protein coding genes (PCGs, CDS) and ribosomal genes (rRNAs) are presented with standard abbreviations. Genes coding for transfer RNAs (tRNAs) are presented with one letter abbreviation. S1 = AGN, S2 = UCN, L1 = CUN, L2 = UUR. Wathet blue, pink, light green and orange represents PCGs, rRNAs, tRNAs and D-loop (non-coding control region), respectively. Black, green and purple represents GC content, positive GC skew (GC skew+) and negative GC skew (GC skew-), respectively. Gene orientation is indicated by arrows.

species [19, 27]. A total of 39 bp intergenic spacers were distributed across eight locations. The shortest intergenic spacers were one bp, and the longest intergenic spacer which was located between *trnS2* and *nad1*, was 21 bp. A total of 28 bp overlaps were distributed across seven locations. The shortest overlaps were one bp, and the longest overlap was eight bp which was located between *trnW* and *trnC*. two seven bp overlaps were located between *atp8* and *atp6*, between *nad4* and *nad4L*.

The mitogenome consisted of 42.26% A, 30.54% T, 16.54% C and 10.65% G (S3 Table). The A+T content of the whole genome, PCGs, tRNAs, rRNAs and non-coding control region was 72.81%, 72.79%, 75.35%, 75.59% and 67.17%, respectively. It seemed that the name of A-T rich region was not suitable for non-coding control region. The AT-skew value was 0.161, and the GC-skew value was -0.220, which indicated that the mitogenome had a preference to A and C. The AT-skew of Coreidae species ranged from 0.097 (*Leptoglossus membranaceus*) to 0.176 (*Acanthocoris* sp. FS-2019), and GC-skew ranged from -0.253 (*Hydaropsis longirostris*) to -0.187 (*Clavigralla tomentosicollis*), which indicated A and C were more abundant than T and G in Coreidae species, respectively.

## Protein-coding genes

The total length of 13 PCGs of *N. montanus* was 11047 bp (S2 Table), accounting for 72.79% of the whole genome, which was medium size in Coreidae species (from 11020 bp to 11110 bp). For all Coreidae species, all AT-skew values were positive, and GC-skew values were positive except for *Leptoglossus membranaceus* (-0.008) and *Notopteryx soror* (-0.001). Four PCGs, including *nad1*, *nad4*, *nad4L*, and *nad5*, were encoded by the N strand, and the remaining nine PCGs were encoded by the J strand (Fig 1 and S2 Table). In *N. montanus*, *cox1* used TTG as start codon, however, the remaining 12 PCGs used ATN as start codon (ATA for *atp8*, *nad3*, and *nad6*; ATC for *cox2*; ATG for *atp6*, *cox1*, *cox3*, *nad2*, *nad4*, *nad5*, and *cytb*; ATT for *nad1*, and *nad4L*). Five PCGs, including *atp8*, *atp6*, *nad5*, *nad4*, and *nad4L*, used TAA as stop codon; however, two PCGs (*nad3*, and *nad6*) and six PCGs (*cox1*, *cox2*, *cox3*, *cytb*, *nad1*, and *nad2*) used TA and T as incomplete stop codon, respectively (S2 Table).

Relative synonymous codon usage (RSCU) analysis of *N. montanus* showed that a total of 62 codons were used, except for two stop codons (UAA and UAG) (S4 Table). The highest frequent four codons were UUA (264), UUU (251), AAU (243), and AUA (204), which accounted for 26.13% total number of codons. These codons were composed of either A or U. On the contrary, high G+C contents codons, including GCG (2), CGC (3), and CGG (4), were the lowest frequent codons. For codons with RSCU values more than one, they were more likely to use either A or T rather than either G or C at the third position of the codon (Fig 2A and 2B). The heatmap of codon usage for 31 Coreoidea species showed that codons used either A or T at the third position of the codon which were more frequently used than these used either G or C.

For amino acids, leucine 2, phenylalanine, isoleucine, methionine, tyrosine, and asparagine, were the most abundant in *N. montanus*, which accounted for 9.92%, 8.92%, 8.86%, 7.52%, 6.96% and 6.15% the total number of amino acids, respectively (Fig 2C). The heatmap of amino acid usage for 31 Coreoidea species showed that leucine 2, phenylalanine, isoleucine and tyrosine were the most abundant, and the abundance variation was small. However, the abundance of methionine and asparagine had larger variation among species than that of leucine 2, phenylalanine, isoleucine and tyrosine. Furthermore, there was a linear correlation between the effective number of codons (ENc) and the G+C content of 13 PCGs (S1 Fig). Specifically, ENc had a better linear correlation with the G+C content of the third position of the codon than the remaining positions.

*Nad5* and *cytb* had the lowest and highest Ks values, respectively (Fig 3). *Cox1* had the lowest value of Ka and Ka/Ks, *atp8* had the highest value of Ka, and *nad5* the highest value of Ka/Ks (Fig 3). The value of Ka/Ks for 13 PCGs was less than 1, which indicated that all PCGs were evolving under purifying selection. There were negative correlations between the value of Ka/Ks and G+C contents (S2 Fig).

## Transfer RNA and ribosomal RNA genes

The total length of 22 tRNAs of *N. montanus* was 1444 bp. The shortest tRNAs were 63 bp (*trnI*, *trnC*, *trnY*, *trnD*, *trnG*, *trnA*, *trnT* and *trnP*), while the longest tRNA was 75 bp (*trnK*). The total A+T content of tRNAs was 75.35%. The AT-skew value was 0.079, and the GC-skew value was -0.096. Eight tRNAs (*trnQ*, *trnC*, *trnY*, *trnF*, *trnH*, *trnP*, *trnL*, and *trnV*) were encoded by the N strand, and the remaining tRNAs were encoded by the J strand (Fig 1 and S2 Table). The distribution pattern of tRNAs in the mitogenome was in accordance with that of other Coreidae species.

For secondary structure, *trnS1* lacked a dihydrouridine (DHU) arm (Fig 4). The remaining tRNAs had a typical cloverleaf secondary structure. Non-Watson-Crick base pairing (G-U

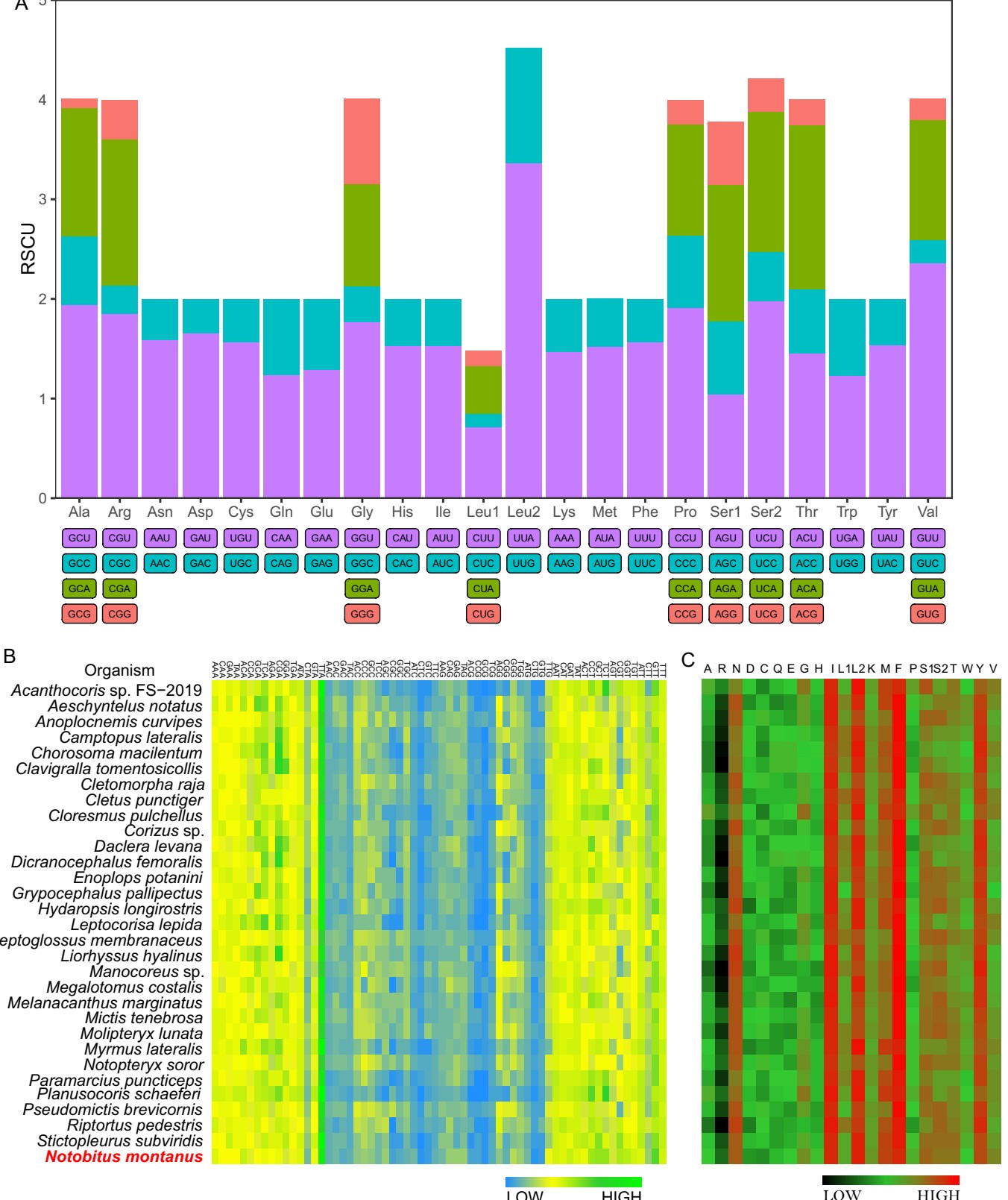

**Fig 2. Codon and amino acid usage.** (A) Relative synonymous codon usage (RSCU) of *N. montanus*. (B) Heatmap indicates codon usage in 31 Coreoidea species. (C) Heatmap indicates amino acid usage in 31 Coreoidea species.

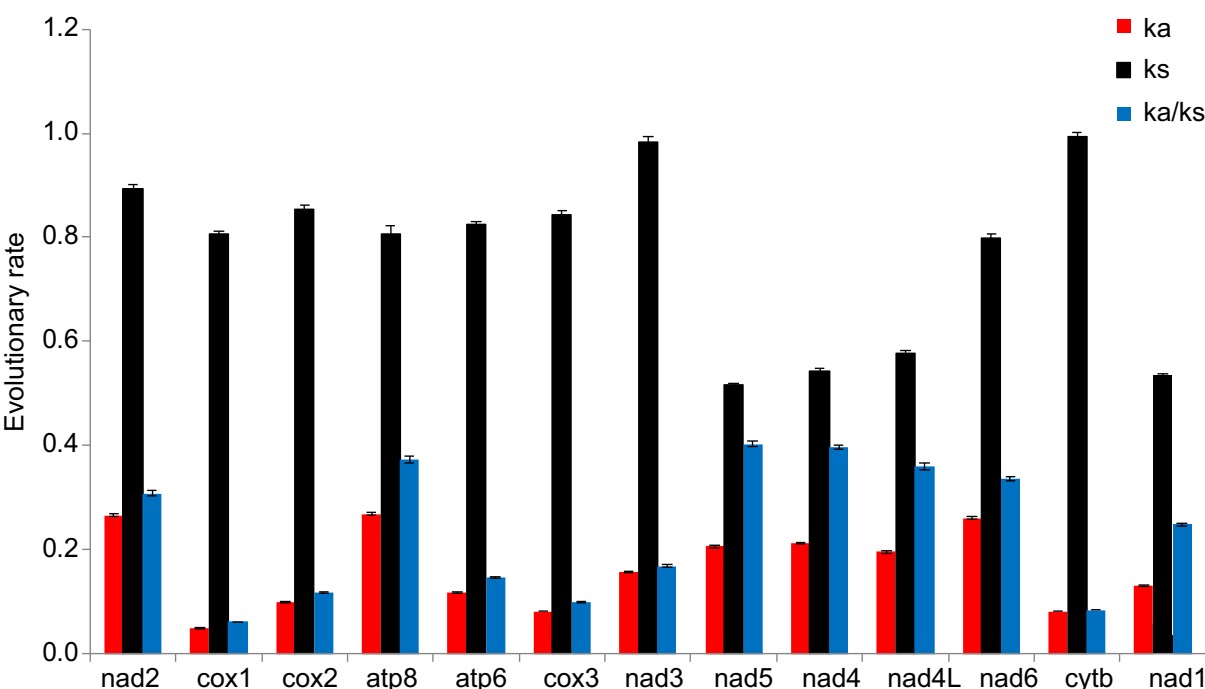

**Fig 3. Evolutionary rates of 13 protein-coding genes in the mitogenome of 31 Coreoidea.** Red represents nonsynonymous nucleotide substitutions per nonsynonymous site (Ka); black represents synonymous nucleotide substitutions per synonymous site (Ks); blue represents the ratio of Ka to Ks (Ka/Ks). Error bars presented the standard error of mean.

pairing) appeared in the acceptor stem of *trnA*, *trnH* and *trnV*. G-U pairing also appeared in the anticodon arm of *trnV*, and the TΨC arm of *trnS1* and *trnV*. Four tRNAs (*trnC*, *trnH*, *trnT*, and *trnS2*) used U as the discriminator nucleotide, and the remaining 18 tRNAs used A as the discriminator nucleotide.

The mitogenome of *N. montanus* encoded two rRNAs. *RrnL* (16s rRNA) was 1281 bp which was located between *trnL1* and *trnV*, and *rrnS* (12s rRNA) was 784 bp which was located between *trnV* and the non-coding control region. Both were encoded by the N strand (S2 Table). The A+T content of rRNAs was 75.59%, the AT-skew value was 0.213, and the GC-skew value was -0.310 (S3 Table).

## Non-coding control region

The non-coding control region (D-loop) of *N. montanus*, located between 12s rRNA and *trnI*, was 1642 bp (Fig 5). Its position in the mitogenome was accordance with that of other Coreidae species [6]. The A+T content of D-loop was 67.17%. AT-skew value was 0.081, and GC-skew value was -0.328 (S3 Table). Among the 15 Coreidae species, the length of D-loop ranged from 794 bp (*Cletus rubidiventris*) to 3441 bp (*C. pulchellus*), which made significant contribution to the size variation of mitogenome (Fig 5) [28]. No tandem repeat was detected in the non-coding region of *N. montanus* (Fig 5). Similar phenomenon was occurred in *Hydaropsis longirostris*, *Pseudomictis tenebrosa*, *Manocoreus* sp. However, most Coreidae species had tandem repeats in their non-coding control region. For Coreidae species, the shortest tandem repeat unit in D-loop was seven bp (*Molipteryx lunata*), and the longest tandem repeat unit was 369 bp (*C. pulchellus*) (Fig 5).

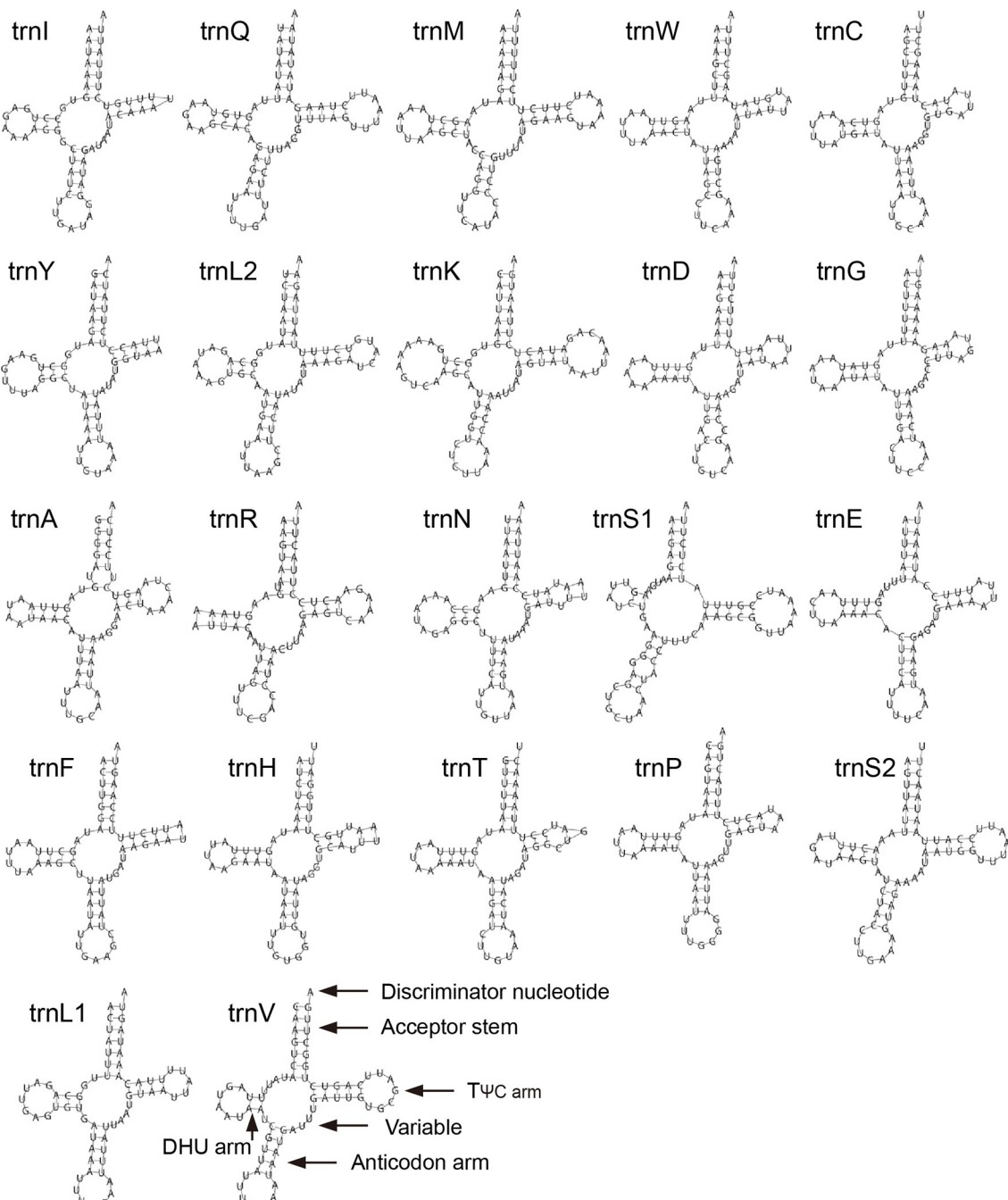

**Fig 4. Predicted secondary structure of 22 tRNAs of *N. montanus*.**

## Phylogenetic relationship

Phylogenetic analyses were carried out with nucleotide sequences of 13 PCGs extracted from 31 mitogenomes of Coreoidea (S1 Table). The topologies of Bayesian inference (BI) and maximum likelihood (ML) phylogenetic trees, with high support scores at most nodes, were identical except for Alydinae (Fig 6 and S3 Fig). Bayesian tree showed that five Alydinae species' relationship was (((*Megalotomus costalis* + *Riptortus pedestris*) + *Camptopus lateralis*) +

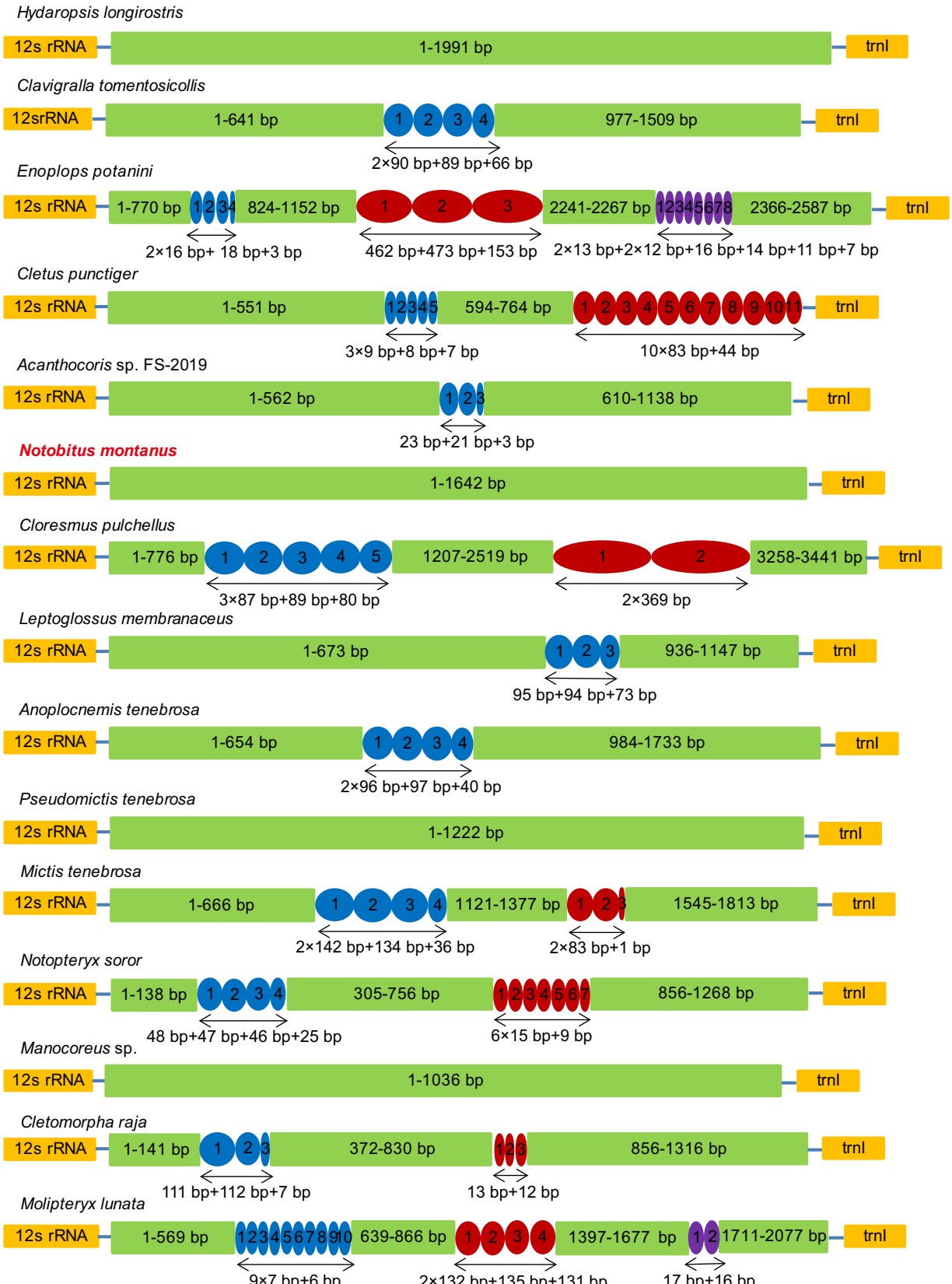

**Fig 5. Tandem repeat sequences in the non-coding control region.** The copy number of tandem repeats are shown by colored (blue, red, purple) oval with Arabic numerals. Non-repeat regions are indicated by green box with sequence size inside.

(*Daclera levana* + *Melanacanthus marginatus*)). However, maximum likelihood tree showed that their relationship was ((((*Camptopus lateralis* + *Riptortus pedestris*) + *Melanacanthus marginatus*) + *Daclera levana*) + *Megalotomus costalis*). Compared with maximum likelihood tree, Bayesian tree had higher support scores.

Stenocephalidae had a sister group relationship with the remaining families of Coreoidea (PP = 1, BS = 98.60). Stenocephalidae and Rhopalidae, were monophyly; however, Alydidae and Coreidae were not monophyly (Fig 6). At subfamily level, Alydinae and Coreinae showed monophyly with high support (PP = 1, BS>97). Their relationship was (((((Alydinae + Pseudophloeinae) + Hydarinae) + Micrelytrinae) + Coreinae) + Rhopalinae). At present, Coreinae have 13 genera with complete mitogenomic information, their relationship was (*Manocoreus* + (*Enoplops* + *Cletus*)) + (((*Acanthocoris* + (*Notobitus* + *Cloresmus*)) + (*Leptoglossus* + (*Mictis* + (*Pseudomictis* + (*Anoplocnemis* + (*Molipteryx* + (*Notopteryx* + *Cletomorpha*)))))))).

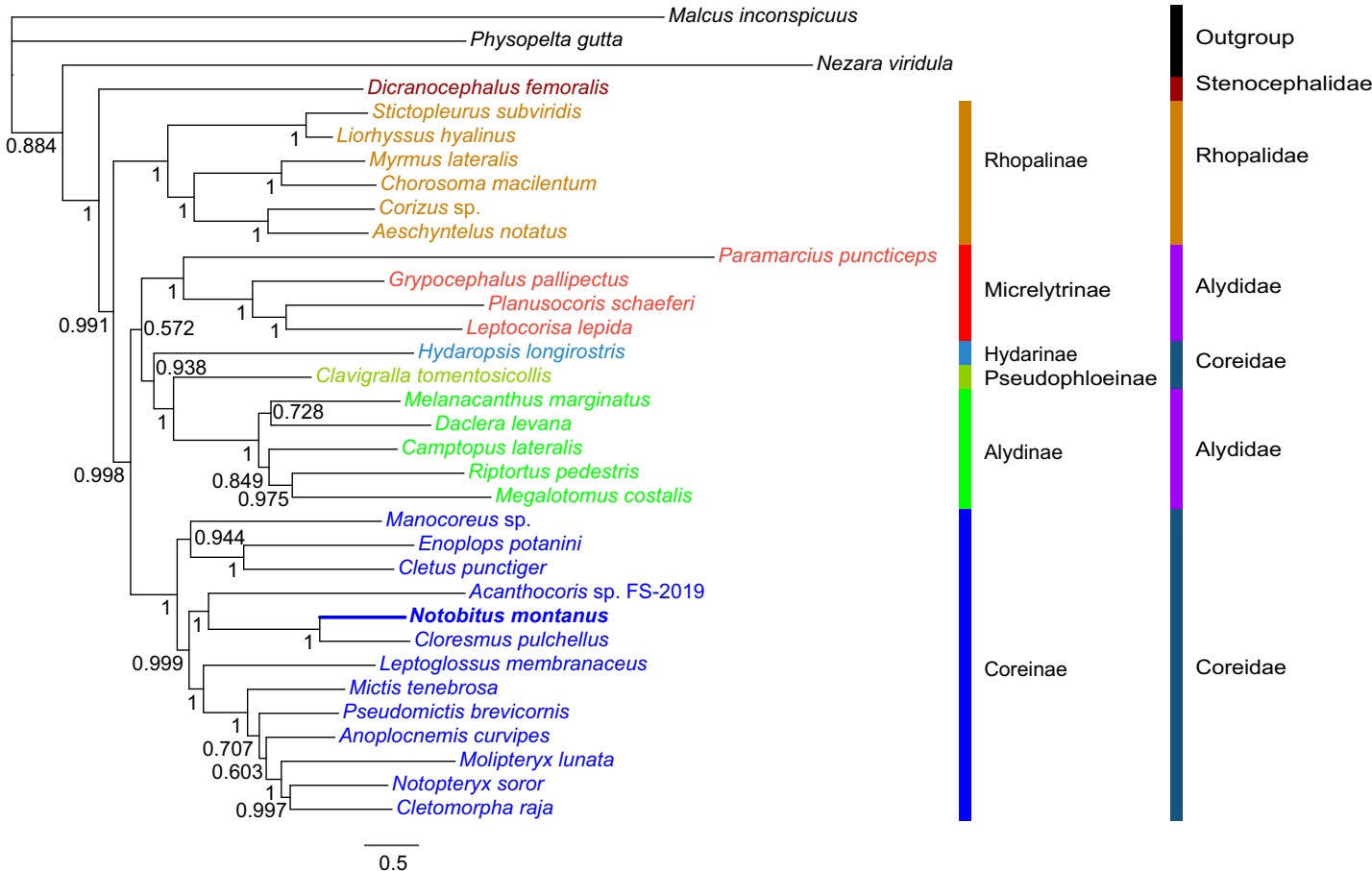

**Fig 6. Bayesian phylogenetic tree of 31 Coreoidea species.** The tree is constructed with nucleotide sequences of 13 mitochondrial PCGs. For family, Black, red, orange, purple and dark blue represents outgroup, Stenocephalidae, Rhopalidae, Alydidae and Coreidae, respectively. For subfamily, orange, bright red, wathet blue, green, bright green, and bright blue represents Rhopalinae, Micrelytrinae, Hydarinae, Pseudophloeinae, Alydinae, and Coreinae, respectively. The posterior probabilities were labeled at each node.

## Discussion

Two seven bp overlaps were detected in PCGs, including between *atp8* and *atp6*, between *nad4* and *nad4L*. These two overlaps are popular in arthropods [14, 15, 29]. The A+T content of PCGs was 75.79%, the AT-skew value was -0.116, and the GC-skew value was -0.004 (S3 Table). It seemed that PCGs had a preference to T than A, and no obvious preference between G and C. In *N. montanus*, only *cox1* used TTG as start codon, and the remaining 12 PCGs used ATN as start codon. Our results are in accordance with that most PCGs use ATN as start codon in metazoan [10]. TTG is usually used as start codon for *cox1* in animals [30, 31]. Eight PCGs used TA (*nad3* and *nad6*) or T (*cox1*, *cox2*, *cox3*, *cytb*, *nad1*, and *nad2*) as incomplete stop codon. Incomplete stop codons are usually used in metazoan, which can be added with either A or AA by post-transcriptional polyadenylation [10, 13]. Relative synonymous codon usage (RSCU) analysis of *N. montanus* showed that they were more likely to use either A or T rather than either G or C at the third position of the codon. For 31 Coreoidea species, they preferred to use codons which used either A or T at the third position of the codon than codons which used either G or C at the third position of the codon. Furthermore, there was a positive correlation between the effective number of codons (ENc) and the G+C content of 13 PCGs (S1 Fig). Taken together, G+C content could influence codon usage [32, 33]. The value of Ka/Ks for 13 PCGs was less than 1, which indicated that all PCGs were evolving under purifying selection. As a result, all PCGs could be used to construct phylogenetic tree. *Nad5* had the highest value of Ka/Ks, which suggested that *nad5* had the fastest evolutionary rate in all PCGs. *Cox1*, with the lowest value of Ka/Ks, had the strongest purifying selection stress. *Cox1* is suitable for barcoding marker. Correlation analysis indicated that the value of Ka/Ks had a negative correlation with G+C content (S2 Fig). In short, G+C content not only influences codon usage but also evolutionary rate.

The AT-skew value was 0.079, and the GC-skew value was -0.096, which indicated that tRNAs had a weak preference to A and C. Most tRNAs had the typical cloverleaf secondary structure, but *trnS1* lacked a dihydrouridine (DHU) arm. Previous studies reveal that the phenomenon of *trnS1* lacking DHU arm is popular in insects [34]. Four tRNAs (*trnC*, *trnH*, *trnT*, and *trnS2*) used U as the discriminator nucleotide, and the remaining 18 tRNAs used A as the discriminator nucleotide. This phenomenon occurs in other insects, such as *Dinorhynchus dybowskyi* (NC_037724) [29]. Non-Watson-Crick base pairing (G-U pairing) appeared in the acceptor stem, anticodon arm and TΨC arm of tRNAs. Non-Watson-Crick base pairing in tRNA could increase the stability of tRNAs, which is popular in insects [13]. The A+T content of rRNAs was 75.59%, the AT-skew value was 0.213, and the GC-skew value was -0.310 (S3 Table). It seemed that rRNAs had similar A+T content with tRNAs, however, rRNAs had stronger preference to A and C than tRNAs.

The position of non-coding control region (D-loop) in *N. montanus* was accordance with that of other Coreidae species [6]. The A+T content of D-loop was 67.17%, which was lower than PCGs, tRNAs and rRNAs. It seems that the name of A-T rich region is not suitable for non-coding control region. AT-skew value was 0.081, and GC-skew value was -0.328 (S3 Table), which suggested that non-coding control region had a weak preference to A but had a strong preference to C. The length of non-coding control region ranged from 794 bp (*Cletus rubidiventris*) to 3441 bp (*C. pulchellus*) in 15 Coreidae species, which made significant contribution to the size variation of mitogenome (Fig 5) [28]. No tandem repeat was detected in the non-coding region of *N. montanus* (Fig 5). Similar phenomenon was occurred in *Hydaropsis longirostris*, *Pseudomictis tenebrosa*, *Manocoreus* sp [34]. Most Coreidae species had tandem repeats in their non-coding control region. For Coreidae species, the shortest tandem repeat unit was seven bp (*Molipteryx lunata*), and the longest tandem repeat unit was 369 bp (*C.

*pulchellus*) (Fig 5), which might influence mitochondrial replication and post-transcriptional modifications [10, 13]. High frequent mutations were occurred in the D-loop, making region was variable in both size and nucleotide sequence [28].

Coreoidea consists of five extant families: Coreidae, Alydidae, Rhopalidae, Stenocephalidae and Hyocephalidae [1, 6, 7]. Currently, no mitogenome of Hyocephalidae species is available in NCBI, as a result, only four families were analyzed in the study. Stenocephalidae had a sister group relationship with the remaining families of Coreoidea (PP = 1, BS = 98.60), which was consistent with the previous study [6]. Stenocephalidae and Rhopalidae, were monophyly; however, Alydidae and Coreidae were not monophyly (Fig 6), which was in accordance with previous studies [1, 6].

At subfamily level, Alydinae and Coreinae showed monophyly with high support (PP = 1, BS>97). Our phylogenetic tree, based on 13 PCGs, showed that their relationship was (((((Alydinae + Pseudophloeinae) + Hydarinae) + Micrelytrinae) + Coreinae) + Rhopalinae). One study, based on 13 PCGs and 2 rRNAs showed their relationship was ((((Coreinae + Hydarinae) + Micrelytrinae) + (Alydinae + Pseudophloeinae)) + Rhopalinae) [6]. Another study, based on ultraconserved element loci showed their relationship was ((((Micrelytrinae + Hydarinae) + (Alydinae + Pseudophloeinae)) + Coreinae) + Rhopalinae) [1]. Rhopalinae was a sister group of the remaining five subfamilies in common. In addition, Alydinae had a closer relationship with Pseudophloeinae than the remaining subfamilies. Our tree showed that Coreinae was a sister group of the remaining four subfamilies, including Alydinae, Pseudophloeinae, Hydarinae, Micrelytrinae, which was consistent with the tree based on ultraconserved element loci but not the tree based on 13 PCGs and 2 rRNAs. Previous study indicated that the sequence of rRNAs was not suitable for phylogenetic tree reconstruction [34]. However, the topology of our tree was in accordance with that of tree based on both mitochondrial and nuclear genes [35] but not ultraconserved element loci [1]. It seems that their relationships are complex, and their relationships need revaluation and revision [9].

At present, Coreinae have 13 genera with complete mitogenomic information, their relationship was (*Manocoreus* + (*Enoplops* + *Cletus*)) + (((*Acanthocoris* + (*Notobitus* + *Cloresmus*)) + (*Leptoglossus* + (*Mictis* + (*Pseudomictis* + (*Anoplocnemis* + (*Molipteryx* + (*Notopteryx* + *Cletomorpha*)))))))). Coreinae, with 372 genera, is the largest subfamily of Coreidae. The mitogenome of most genera remains unknown. Owing to a large number of species belonging to Coreinae (2320 species), phylogenetic analysis with ultraconserved elements showed that some genera were not monophyletic [9], which suggested that the relationships among Coreinae species were complex, and need further study. The mitogenomic information of *N. montanus* could shed light on the evolution of Coreoidea.

## Conclusions

This study has decoded the complete mitogenome of *N. montanus*. Codon usage analysis indicated that high frequently used codons used either A or T at the third position of the codon. Amino acid usage analysis showed that leucine 2, phenylalanine, isoleucine and tyrosine were the most abundant in 31 Coreoidea species. The value of Ka/Ks for 13 PCGs was less than one, which suggested that all PCGs were evolving under purifying selection, *Nad5* and *Cox1* had the lowest and strongest purifying selection stress, respectively. Correlation analysis showed that A+T content could influence codon usage and evolutionary rate. The phylogenetic tree showed that Alydidae and Coreidae were not monophyletic. However, the topology of phylogenetic trees, based on 13 PCGs, was in accordance that of tree based on both mitochondrial and nuclear genes but not ultraconserved element loci or combination of 13 PCGs and two rRNAs. It seems that their relationships are complex, and their relationships need revaluation

and revision. The mitogenomic information of *N. montanus* could shed light on the evolution of Coreoidea.

## Supporting information

**S1 Fig. Effective numbers of codons have a positive correlation with the G+C content of codons.** (A) total GC content of codons; (B) GC content of the first position of the codon; (C) GC content of the second position of the codon; (D) GC content of the third position of the codon.
(EPS)

**S2 Fig. The Ka/Ks has a negative correlation with the G+C content of codons.** (A) total GC content of codons; (B) GC content of the first position of the codon; (C) GC content of the second position of the codon; (D) GC content of the third position of the codon.
(EPS)

**S3 Fig. Maximum likelihood phylogenetic tree of 31 Coreoidea species.** The tree is constructed with nucleotide sequences of 13 mitochondrial PCGs. For family, Black, red, orange, purple and dark blue represents outgroup, Stenocephalidae, Rhopalidae, Alydidae and Coreidae, respectively. For subfamily, orange, bright red, wathet blue, green, bright green, and bright blue represents Rhopalinae, Micrelytrinae, Hydarinae, Pseudophloeinae, Alydinae, and Coreinae, respectively. The bootstrap values were labeled at each node.
(EPS)

**S1 Table. List of species for phylogenetic analysis.**
(DOCX)

**S2 Table. Annotation of the *Notobitus montanus* mitogenome.**
(DOCX)

**S3 Table. Nucleotide composition of *Notobitus montanus* (%).**
(DOCX)

**S4 Table. Codon usage in the mitochondrial genome of *Notobitus montanus*.**
(DOCX)

## Author Contributions

**Conceptualization:** Qianquan Chen.

**Data curation:** Xiaoke Tian, Yongqin Li, Qin Chen, Qianquan Chen.

**Formal analysis:** Xiaoke Tian, Qianquan Chen.

**Funding acquisition:** Qianquan Chen.

**Investigation:** Xiaoke Tian, Yongqin Li, Qin Chen, Qianquan Chen.

**Methodology:** Qianquan Chen.

**Project administration:** Qianquan Chen.

**Resources:** Qianquan Chen.

**Software:** Xiaoke Tian, Qianquan Chen.

**Supervision:** Qianquan Chen.

**Validation:** Xiaoke Tian, Qianquan Chen.

**Visualization:** Xiaoke Tian, Qianquan Chen.

**Writing – original draft:** Xiaoke Tian, Qianquan Chen.

**Writing – review & editing:** Qianquan Chen.

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
