## [Decision Letter · Decision Letter 0]

28 Sep 2022

PONE-D-22-21262Mitogenome of Notobitus montanus (Hemiptera: Coreidae) and a phylogenetic analysis of CoreoideaPLOS ONE

Dear Dr. Chen,

Thank you for submitting your manuscript to PLOS ONE. After careful consideration, we feel that it has merit but does not fully meet PLOS ONE’s publication criteria as it currently stands. Therefore, we invite you to submit a revised version of the manuscript that addresses the points raised during the review process.

Both the reviewer and the academic editor have found your manuscript interesting. However, both have comments on manuscript preparation and authors will have to revise the manuscript substantially based on the comments provided at the bottom of this email. 

We look forward to receiving your revised manuscript.

Kind regards,

Neelesh Dahanukar, Ph.D.

Academic Editor

PLOS ONE

Journal Requirements:

"This research was funded by National Natural Science Foundation of China, grant number 32060124; Guizhou Normal University, grant number Qianshixinmiao[2021]A11"

"The authors declare no conflicts of interest."

Additional Editor Comments:

Authors have provided complete mitogenome of the Leaf-footed bug Notobitus montanus and its analysis. Apart from routine analysis, authors have also provided some comparative genomics and phylogenetics. Although the study is interesting and the analysis is performed appropriately, the manuscript is not presented well and authors will have to revise it carefully before a final decision can be made. I have made the concerns clear below.

Major comment

In several cases authors draw major conclusions without providing data or analysis. Most of this conclusion could also be because authors are not aware of several basic concepts in molecular genetics of mitochondria. Authors should know that, while making any big statement they should provide evidence or arguments based on earlier studies to support the claims.

The first claim authors make is that “Most amino acids preferred to use tRNAs encoded by nuclear genome.” It is unclear how authors came to this conclusion. The tRNAs for all 20 amino acids (including the two separate starting codons for leucine and serine) are present in the mitochondrial genome so why do authors mention that the mitochondria is using nuclear tRNAs. If this conclusion is drawn based on the mismatch between the third codon position in the CDS and tRNA anticodon arm, then authors should know that the third codon position is a wobble position.

The second claim authors make is regarding the overlap between the genes. Authors mention, “The presence of polycistron might contribute to generating overlaps between PCGs.” Authors cite reference [32] for this statement; however, the reference does not make any such claim. This statement is an ignorance of basic molecular biology of mitochondrial genome. First, polycistron has nothing to do with overlap in the genes. Most polycistron exists without overlap in the genes. In the case of mitochondria, the mitochondrial genes are generally transcribed as two large precursor polycistronic transcripts. These transcripts are subsequently cleaved to generate individual mRNAs, tRNAs and rRNAs. So polycistron does not explain the overlapping genes because even the genes that do not have overlap are transcribed from polycistronic transcripts.

Minor comments: Authors will need to improve manuscript writing substantially.

(1) Authors can add the common name “Leaf-footed bug” before Notobitus montanus to make it more accessible to non technical readers. Suggested title: Mitogenome of the leaf-footed bug Notobitus montanus (Hemiptera: Coreidae) and a phylogenetic analysis of Coreoidea

(2) As per the zoological nomenclatural rules, authority of scientific names that are written as per their original combination are not in parenthesis. Notobitus montanus is the original combination in which the species was described. So the authority and year cannot be in parenthesis. This should be - Notobitus montanus Hsiao, 1963. Make this change in abstract and remaining text.

(3) Abstract is not drafted properly. Authors unnecessarily provide details on the start and stop codons of coding genes when most of this is common in other organisms as well. In the manuscript authors talk about RSCU, compare codon usage with other species, they also provide comparison of D-loop of several species but none of this analysis is mentioned in the abstract. An abstract is a vital part of the manuscript and should be presented properly with a proper flow. There should be one or two lines providing the background of the study and its importance. One or two lines describing the methods use (also mention you determined secondary structure of tRNAs, compared genomes for codon usage and performed phylogeny). Important results in a few lines and implications of this study in understanding ecology and evolution of the species as a conclusion.

(4) The word “taxology” is normally not used in the field. What authors are referring to is called “taxonomy”.

(5) Introduction does not provide any information on the taxa under study Notobitus montanus and why it is important to study its mitogenome.

(6) As a rule, authors should use present tense when they are referring to established facts published earlier or any other facts. For example, second last line in first paragraph of results and discussion, “These two overlaps were popular in arthropods [19, 20]” should be “……. overlaps are popular in arthropods…”. Last line in third paragraph of results and discussion, “Incomplete stop codons were usually used in metazoan [18]” should be “Incomplete stop codons are usually used in metazoan [18]”. Similarly, in figure caption of figure 6, “The posterior probabilities were labeled at each node” should be “The posterior probabilities are labeled at each node”.

(7) The results are not stated properly and discussed at length. It is actually better to separate results and discussions. Authors can explain all the results in the results section and in the discussion discuss the results with respect to other mitogenomes and also discuss the implications of the results with respect to understanding ecology and evolution of the focal taxa. Authors have provided some analysis with respect to the published genomes of related taxa but the implications of the results are not clear.

(8) Figure legends should be self-explanatory and readers should understand the figure from figure legends without referring to the main text. For example, authors use several colours, numbers, etc. in figure 5 but the figure legend does not explain any of these. Similarly, for figure 6, there are two classification schemes shown in the figure but these are not explained in the figure legend. Figure legend of figure 6 also does not explain what data was used for the phylogenetic analysis, both coding and non-coding genes or just the coding genes. Although this is mentioned in the main text, as stated earlier, figure legends should be in details and self-explanatory.

(9) Supplementary information should be referred to as Fig. S1, Fig. S2, Table S1, etc. and not S1 Fig., S2 Fig., S1 Table, etc. Also, revise the figure captions to make them self-explanatory.

Reviewers' comments:

Reviewer's Responses to Questions

**Comments to the Author**

1. Is the manuscript technically sound, and do the data support the conclusions?

Reviewer #1: Yes

2. Has the statistical analysis been performed appropriately and rigorously? 

Reviewer #1: Yes

3. Have the authors made all data underlying the findings in their manuscript fully available?

Reviewer #1: Yes

4. Is the manuscript presented in an intelligible fashion and written in standard English?

Reviewer #1: Yes

5. Review Comments to the Author

Reviewer #1: Dear Editor,

This manuscript focuses on sequencing a mitogenome of Notobitus montanus (Hemiptera: Coreidae) - the first mitogenome of Notobitus to investigate the mitogenome structure and phylogeny analysis. Their results were consistent with Dong et al. 2022 and didn't get any innovative results but one mitogenome of a species. I have some key points that I will address here if you think it was deserved to publication.

Reference

Dong X, Wang K, Tang Z, Zhang Y, Yi W, Xue H, et al. Phylogeny of Coreoidea based on mitochondrial genomes show the paraphyly of Coreidae and Alydidae. Archives of Insect Biochemistry and Physiology. 2022;110(1):e21878.

Comments to Author:

Major questions:

Q1: Coreoidea consists of five extant families, but mitochondrial genomes in your manuscript only covered four families (Hyocephalidae not included). You should state clearly the group used for phylogeny reconstruction in all related description. Such as this sentence “The phylogenetic tree of Coreoidea was constructed with the new mitogenome and 30 mitogenomes extracted from NCBI.”

Q2: The methods for genome extraction, sequencing and mitochondrial genome assembly are inaccurate in materials and methods. “DNA was isolated from…. Mitogenomic DNA was fragmented, then ~500 bp DNA was recycled. Paired-end libraries were constructed with the Illumina platform. The mitogenome was sequenced using the…”. It confused me for obtaining the mitogenome between genomic DNA extraction and sequencing or assembly. There is more than one way to get to the mitogenome (sequence genomic DNA and then to isolate mitogenome after sequencing; enrich the mitogenome prior to sequencing and other approach).

Q3. In the Transfer RNA and ribosomal RNA genes results, you described the amino acid preferences about the use of tRNAs. “…….preferred to use tRNAs encoded by the mitogenome or nuclear genome”. How do you determine the use in the nuclear genome? And you operate only for Notobitus montanus, cannot represent the whole large group, whether there are relevant references to justify?

Q4: I think that authors should give a detailed and accurate description in results and discussion part about phylogenetic relationship. For example, you cited the reference but inaccurate. “However, based on nucleotide sequences of 13 PCGs and two rRNAs, some studies considered Alydidae and Coreidae were monophyletic [40].” It clarified the sister group of Rhopalidae with Alydidae + Coreidae but not revealed the monophyly of Alydidae and Coreidae because one subfamily of Alydidae were not covered.

Minor comments:

Q1. You used many analyses software in materials and methods, but not all of them has corresponding version number. Determine the software version you are using.

Q2. In phylogenetic analysis part, I don’t understand the operation for PCGs in Phylosuite. “These mitogenomes were imported into PhyloSuite. After standardization of sequences, the nucleic acid sequences of 13 PCGs were extracted from these mitogenomes.” I wonder what’s meaning and how to standardize in Phylosuite. That is, what is being done to the sequences.

6. PLOS authors have the option to publish the peer review history of their article (what does this mean?). If published, this will include your full peer review and any attached files.

Reviewer #1: No

---

## [Author Response · Author response to Decision Letter 0]

13 Nov 2022

We have carefully read the comments from the editors and reviewers. The comments and suggestions are very valuable for us to improve this manuscript. Please see manuscript and Response to Reviewers for specific revision!

---

## [Editor Report · Decision Letter 1]

22 Dec 2022

PONE-D-22-21262R1Mitogenome of the leaf-footed bug Notobitus montanus (Hemiptera: Coreidae) and a phylogenetic analysis of CoreoideaPLOS ONE

Dear Dr. Chen,

Thank you for submitting your manuscript to PLOS ONE. After careful consideration, we feel that it has merit but does not fully meet PLOS ONE’s publication criteria as it currently stands. Therefore, we invite you to submit a revised version of the manuscript that addresses the points raised during the review process. Revised manuscript is a substantial improvement over the earlier draft. However, there are some minor issues that authors will have to resolve before the manuscript can be finally accepted. Specific comments are made in the attached manuscript file.

We look forward to receiving your revised manuscript.

Kind regards,

Neelesh Dahanukar, Ph.D.

Academic Editor

PLOS ONE

Journal Requirements:

Additional Editor Comments:

Revised manuscript is a substantial improvement over the earlier draft. However, there are some minor issues that authors will have to resolve before the manuscript can be finally accepted. Specific comments are made in the attached manuscript file.
---

## [Author Response · Author response to Decision Letter 1]

9 Jan 2023

We have carefully read the comments from the editor and revised the manuscript according to the comments.

---

## [Editor Report · Decision Letter 2]

27 Jan 2023

Mitogenome of the leaf-footed bug Notobitus montanus (Hemiptera: Coreidae) and a phylogenetic analysis of Coreoidea

PONE-D-22-21262R2

Dear Dr. Chen,

We’re pleased to inform you that your manuscript has been judged scientifically suitable for publication and will be formally accepted for publication once it meets all outstanding technical requirements.

Kind regards,

Neelesh Dahanukar, Ph.D.

Academic Editor

PLOS ONE

---

## [Editor Report · Acceptance letter]

1 Feb 2023

PONE-D-22-21262R2 

Mitogenome of the leaf-footed bug *Notobitus montanus* (Hemiptera: Coreidae) and a phylogenetic analysis of Coreoidea 

Dear Dr. Chen:

I'm pleased to inform you that your manuscript has been deemed suitable for publication in PLOS ONE. Congratulations! Your manuscript is now with our production department. 

Kind regards, 

on behalf of

Dr. Neelesh Dahanukar 

Academic Editor

PLOS ONE